# Thaumatin-like Proteins in Legumes: Functions and Potential Applications—A Review

**DOI:** 10.3390/plants13081124

**Published:** 2024-04-17

**Authors:** Lanlan Feng, Shaowei Wei, Yin Li

**Affiliations:** 1Guangdong Key Laboratory of Plant Resources, School of Life Sciences, Sun Yat-sen University, Guangzhou 510275, China; fengll@link.cuhk.edu.hk; 2Institute of Plant Protection, Henan Academy of Agricultural Sciences, Zhengzhou 450002, China; 3Research & Development Institute of Northwestern Polytechnical University in Shenzhen, Shenzhen 518057, China

**Keywords:** legume, TLP, PR-5, biotic stress, abiotic stress

## Abstract

Thaumatin-like proteins (TLPs) comprise a complex and evolutionarily conserved protein family that participates in host defense and several developmental processes in plants, fungi, and animals. Importantly, TLPs are plant host defense proteins that belong to pathogenesis-related family 5 (PR-5), and growing evidence has demonstrated that they are involved in resistance to a variety of fungal diseases in many crop plants, particularly legumes. Nonetheless, the roles and underlying mechanisms of the TLP family in legumes remain unclear. The present review summarizes recent advances related to the classification, structure, and host resistance of legume TLPs to biotic and abiotic stresses; analyzes and predicts possible protein–protein interactions; and presents their roles in phytohormone response, root nodule formation, and symbiosis. The characteristics of TLPs provide them with broad prospects for plant breeding and other uses. Searching for legume TLP genetic resources and functional genes, and further research on their precise function mechanisms are necessary.

## 1. Introduction

Given their sessile lifestyle, plants have developed a highly complex defense system against different threats, including biotic and abiotic stresses [1]. Global environmental deterioration and pathogen invasions can cause enormous harm to agricultural production. In both animals and plants, several fungal and fungal-like diseases have caused some of the most severe die-offs and extinctions ever witnessed in wild species, and this is jeopardizing food security [2].

Plant pathogenic fungi attack a wide range of crops, but controlling fungal diseases with fungicides is cost intensive and comes with detrimental effects on the environment [3]. Fungal infection threats can be seen across many sources of data, including a high risk of biodiversity loss, host extinction, and low food yield and quality [2]. When plants perceive a pathogenic infection, pathogenesis-related (PR) protein genes are significantly upregulated, acting as the first line of plant defense [1,2]. A recent study showed that 19 classes of PR proteins can be distinguished based on structural similarity and functional activity [1,4], of which, thaumatin-like proteins (TLPs)—the homologies of sweet-tasting thaumatin isolated from the plant *Thaumatococcus daniellii*—belong to the PR-5 family [5,6]. Growing evidence shows that TLPs are involved in resistance to a variety of fungal diseases in many species, such as *Gossypium hirsutum* [7], *Gossypium barbadense* [8], *Solanum lycopersicum* [4,9], *Phyllostachys edulis* [10], *Ganoderma lingzhi* [11], *Triticum aestivum* L. [12,13], *Carya cathayensis* [14], *Fragaria ananassa* [15], *Allium sativum* L. [16], *Pinus radiata* [17], *Lentinula edodes* [18], *Camellia sinensis* [19], *Cucumis melo* L. [20], *Avena nuda* [5], *Musa acuminate* [21], *Manihot esculenta* [22], *Vitis amurensis* [23], *Populus szechuanica* [24], *Camellia sinensis* [25], *Piper colubrinum* [26], *Cynanchum komarovii* [27], *Castanea sativa* [28], *Actinidia chinensis* [29], and *Lyophyllum shimeji* [30]. 

Plant TLPs also play diverse roles in abiotic stresses such as drought [31], freezing [32], and salinity [33,34]. Computational approaches such as cis-regulatory analysis, promoter analysis, and co-expressed gene analysis have revealed that TLPs play diverse roles in abiotic stresses [35]. The transcript levels of *TLPs* are significantly altered under different stress conditions. These findings reveal the functional diversity of TLPs throughout the evolution of genes [36].

Twelve thousand years ago, grain legumes played a crucial role in the development of Neolithic agriculture [37]. Legumes are a globally important crop for food, oil, forage, and bioactive components, and are the main source of edible vegetable protein and oil [38,39,40]. Several phytopathogenic diseases are known to limit legume productivity [41,42]. For example, soybean rust [43], soybean stay-green disease [44], soybean red leaf blotch disease [45], pea blight disease [45], peanut rust [46], peanut leaf spot [47], faba bean root rot [48], and other legume pathogenic diseases are caused by multiple pathogens and are responsible for severe yield loss. Reports have also demonstrated that the productivity of legume crops such as soybeans, chickpeas, and peanuts has been inhibited by different abiotic stresses [49,50,51].

Legumes are some of the most important food crops, and as such, studying biotic and abiotic stresses in the TLP families is worthwhile. This review provides a comprehensive discussion of the research progress regarding TLPs in leguminous plants.

## 2. Classification and Characteristics of TLPs

TLPs are widely distributed in plants, animals, and fungi [52]. In plants, the thaumatin-like PR-5 family includes PR-5 proteins; OLPs (osmotin-like proteins); PR-like rash; the PR5-like protein kinase receptor; and permatins such as zeamation in maize, hordomatin in barley, and avematin in oats [27,53]. Despite their significant diversity in plants, the amino acid sequences of TLPs have a well-defined thaumatin family signature (PS00316): G-x-[GF]-x-C-x-T-[GA]-d-C-x(1,2)-[GQ]-x(2,3)-C [28,33,54]. Furthermore, promoter analysis has shown that *TLPs* contain five conserved TFBSs (transcription factor binding sites) called *ASRC*, *CCAF*, *L1BX*, *NCS1*, and *WBXF* [55]. Specifically, *ASRC* and *WBXF* are responsible for pathogen defense, *CCAF* is associated with the circadian clock, *L1BX* is a homeodomain protein recognition motif, and *NCS1* is a nodulin consensus sequence. 

Furthermore, based on their molecular mass, TLPs can be divided into two types. Large (L)-type TLPs range from 21 to 26 kD and contain 16 conserved cysteine residues; most TLPs belong to this type, including TLP-1 to -5 in *Medicago truncatula* [56], Rj4 in *Glycine max* [57], AdTLP in *Arachis diogoi* [58], and CdTLP in *Cassia didymobotrya* [59]. Small (S)-type TLPs have molecular masses ranging from 16 to 17 kD and have only 10 cysteines in conserved positions because of a peptide deletion [28]. All plant TLPs have similar 3D structures within three domains: domain I, containing 11 stranded β-sheets organized as a β-barrel, forming the protein core; domain II, containing an α-helix and a set of disulfide-rich loops; and domain III, containing a β-hairpin and a coil motif, both maintained by a disulfide bond [31]. There is a cleft structure between domains I and II [60], and each domain is stabilized by at least one disulfide bridge connected by up to 16 cysteine residues with a conserved spatial distribution throughout the protein [61]. Below, AhTLP1 (*Arachis hypogaea* TLP1) is used as an example to exhibit the general legume TLP structure (Figure 1).

In addition to similar core domains, one domain is often responsible for localization at either end of the TLP. The N-terminal signal peptide always targets mature proteins in the secretory pathway. For instance, CkTLP in *Cynanchum komarovii* [27] and AdTLP [58] are located in the extracellular space or cell wall. However, the C-terminal peptide diversifies the subcellular localization of TLPs. For example, the C-terminal propeptide of tobacco osmotin is responsible for its vacuolar localization [62]. Rj4 is presumed to be localized in cell membranes because of a unique transmembrane domain in its C-terminus [57].

## 3. Biological Functions of TLPs in Legumes

TLPs in plants have various biological functions including host–pathogen interactions, stress tolerance, and cell-signaling transduction. They mostly participate in responses to biotic and abiotic stresses [48]. Therefore, it is believed that there are many plant TLPs, and several reports confirm this belief. In total, there are 106 TLPs in *Zea mays*, 97 TLPs in *Gossypium hirsutum*, 78 TLPs in *Oryza sativa*, 66 TLPs in *Nicotiana tabacum*, and 51 TLPs in *Arabidopsis thaliana* [48]. In this regard, research on legumes is still quite limited. The legume genome is extremely complex, and several TLPs have distinct differences. For example, 56 TLPs have been found in *Medicago truncatula*, and only a few MtTLPs have been well studied. However, although research is lacking, a recent study suggests that MtTLPs exhibit high antifungal activity [41].

### 3.1. Legume TLPs in Response to Biotic Stresses

Legume TLPs play important roles in plant defense against various biotic stresses. In 1999, a legume TLP was first isolated from the French bean (*Phaseolus vulgaris* L.), and it exerted antifungal activity against *Fusarium oxysporum*, *Pleurotus ostreatus*, and *Coprinus comatus* [63]. In recent years, more and more TLPs isolated from plants have been proven to have potential antifungal activities [28]. In legumes, the CdTLP protein significantly inhibits fungal strains such as *Candida albicans*, *Candida krusei*, and *Candida parapsilosis* [59]. AdTLP has strong antifungal activity, and transgenic tobacco plants expressing *AdTLP* have shown enhanced resistance to the soil-borne pathogen *Rhizoctonia solani* [58]. *TLP* transcript accumulation was detected after inoculating *Medicago truncatula* with *Colletotrichum trifolii* and *Erysiphe pisi* [64], and recent experimental evidence shows that MtTLP1-5 have strong in vitro antifungal activities against *Rhizoctonia solani*, *Alternaria alternata*, *Fusarium graminearum*, *Fusarium solani*, *Verticillium* sp., and *Phytophthora* spp.; all five MtTLPs reduce the viability of fungal hyphae and significantly reduce *A. alternata* spore germination [56]. Chickpea (*Cicer arietinum*) cells have been shown to experience a rapid response in their activated defense-related genes, including a *TLP* gene, after treatment with a Pmg (*Phytophthora megasperma*) elicitor and cantharidin [65]. In addition, bulked segregant analysis (BSA) has shown that chickpeas also express *TLPs* for *Fusarium oxysporum* resistance [66]. The roles of TLPs in different legumes are summarized in Table 1.

Some TLPs have glucan-binding and glucanase activity, inhibition activity similar to xylanase, α-amylase, or trypsin. Therefore, they have the potential to weaken the cell walls of fungi and interfere with the metabolic processes of pathogens or even affect the digestive ability of insects [28,67,68]. As mentioned above, TLPs contain a conserved motif, and most plant TLPs are the L-type, with 16 highly conserved cysteine residues. Significantly, the 3D structures of TLPs consist of three domains, of which, domains I and II possess a special cleft structure that is crucial for receptor binding and antifungal activity [60]. Specifically, the cleft between domains I and II is made up of five evolutionarily conserved amino acids (arginine, glutamic acid, and three aspartic acid residues) and provides an acidic environment for ligand/receptor binding [61]. Further analysis of this protein superfamily is still necessary to reveal its underlying pathogen-resistance mechanism.

PR proteins accumulate during the plant’s inducible immunity process under pathogenic infection conditions [69]. Pathogens attack plants and trigger innate immune responses, including pattern-triggered immunity (PTI) and effector-triggered immunity (ETI) [70]. Recently, research has shown that host-defense peptides (HDPs) have both direct antimicrobial and immunomodulatory activity, thus protecting multicellular eukaryotes from infections. Importantly, two TLPs—sweet potato LbACP (*Lpomea batatas* anti-cancer peptide) and European plum (*Prunus domestica* L.) PdPR5-1—were recently discovered to be HDPs, and they may provide us with valuable tools for developing phytosanitary products [71]. Regarding legumes, the peanut gene *TLP1b* is a candidate resistant gene that may be used to impart an immune response to *Aspergillus flavus* infections [72]. However, little is known about legume TLPs in terms of plant immune responses. Future research should clarify this subject.

### 3.2. Legume TLPs in Response to Abiotic Stresses

Legume TLPs also protect plants from various abiotic stresses besides biotic invasions. Research has reported that two TLPs in the faba bean (*Vicia faba* L.), VfTLP4-3 and VfTLP5, play important roles in mediating the drought response. These two genes are significantly upregulated under drought conditions, and confer drought resistance and higher POD activity when ectopically expressed in tobacco [36]. Soybean osmotins such as GmOLPa, GmOLPb, and P21e are involved in high-salt stress [33,34]. GmOLPa-like and P21-like osmotins play important roles in drought tolerance; the former shows higher expression levels in roots, and both are also expressed in nodules. Their highest expression levels can be found in the leaves or roots of the drought-tolerant cultivar [31]. The *TLP1b* gene can be induced by wounding the Caesalpinioideae *Senna tora* [73]. The AHCSP33 cold shock protein is a TLP homologous protein secreted into the leaf apoplasts of peanuts (*Arachis hypogaea*) during low-temperature exposure. AHCSP33 acts as a cryoprotecting protein and can prevent the freeze-induced denaturation of _L_-lactate dehydrogenase (LDH) [32]. In addition, a rare evergreen broad-leaved leguminous shrub, *Ammopiptanthus nanus*, expresses large quantities of TLPs to resist low-temperature stress in inhospitable desert areas. Most *AnTLP* genes contain multiple cis-acting elements in promoter regions related to the environmental stress response, and when heterologously expressed in *Escherichia coli*, yeast cells, and tobacco leaves, a cold-induced *AnTLP13* can better enhance low-temperature stress tolerance compared with control cells or seedlings [74]. Moreover, a study on cowpeas (*Vigna unguiculata*) with a supply of excess manganese (Mn) showed a significant increase in soluble apoplastic proteins, including TLPs, in the apoplastic washing fluid of leaf tissue, suggesting that these proteins have specific physiological roles in response to Mn stress [75]. The roles of TLPs resistant to abiotic stress in different legumes are summarized in Table 1. 

**Table 1 plants-13-01124-t001:** The function of legume TLPs in response to stresses.

Name	Sources	Function/Biological Role	References
TLP	*Phaseolus vulgaris*	Antifungal activity	[63]
CdTLP	*Cassia didymobotrya*	Antifungal activity	[59]
AdTLP	*Arachis diogoi*	Antifungal activity	[58]
MtTLP	*Medicago truncatula*	Antifungal activity	[56,64]
TLP	*Cicer arietinum*	Phytoalexin response	[65,66]
VfTLP4-3, VfTLP5	*Vicia faba*	Drought response	[36]
StTLP1b	*Senna tora*	Wound response	[73]
AHCSP33	*Arachis hypogaea*	Cold response	[32]
AnTLPs	*Ammopiptanthus nanus*	Cold response	[74]
GmOLPa	*Glycine max*	Salt response	[33,34]
GmOLPb	*Glycine max*	Salt response	[33,34]
TLP	*Vigna unguiculata*	Manganese toxicity response	[75]
TLP1b	*Arachis hypogaea*	Immune response	[72]
GmOLPa-like and P21-like osmotins	*Glycine max*	Drought response	[31]

### 3.3. The Role of TLPs in Phytohormone Responses

Several expression analyses have shown that hormone treatments have an induction effect on legume *TLP* gene transcript levels, suggesting that TLPs may have a role in different hormone responses with significant crosstalk [34,58,76,77]. Cis-acting elements are commonly identified in the promoter region of *TLP* genes, including motifs controlling phytohormone responses such as auxin, abscisic acid (ABA), ethylene (ET), salicylic acid (SA), jasmonic acid (JA), and gibberellic acid (GA) [74,78,79]. However, there have been few reports on how TLPs act in hormone signal transduction pathways, and there are even fewer regarding legumes. One study showed that two osmotin-like proteins bind cytokinin and its analogs, so they are called cytokinin-binding proteins [80]. It has been suggested that TLPs might be involved in cytokinin transportation partly because of their extracellular locations [58,74].

SA and JA are two important defense hormones in plants. SA is generally involved in defense responses against (hemi-)biotrophic pathogens, while JA is associated with defense responses against necrotrophic pathogens and herbivorous insects; therefore, the SA and JA defense pathways are usually antagonistic [78,79]. *PR5* is activated as an SA response gene during the plant–pathogen interaction process [81]. In leguminous *TLP* genes, SA and JA are spatiotemporally specific to their induction location, and even the same gene usually has opposite expression patterns in a single tissue when faced with these two hormonal stimuli. In soybeans, SA stimulation induces two acidic *PR-5* genes, *GmOLPa* and *P21e*, notably increasing *GmOLPa* levels in the lower leaves. MeJA markedly induces not only neutral *GmOLPb* but also *P21e* in these leaves, but *GmOLPa* transcription levels will then be lower [33]. *PR-5* genes are SA-responsive genes and respond more strongly to SA treatments than JA treatments in *Medicago truncatula*, even though they have similar expression patterns in response to both hormones [82]. *TLP1bs* in *Senna tora* are marker genes for SA signaling and can be induced via wounds: they appear after 6 h and peak after 24 h. However, JA biosynthesis genes can be strongly induced after 1 h and decrease after 3 h, which is unlike that observed in undamaged leaves. This suggests that JA can affect *TLP1b* transcription [73]. Generally, *TLP* genes are more active in the SA pathway, and their activities can be affected by protein surface pH levels and other changes. However, more evidence is needed to support these hypotheses (Figure 2).

### 3.4. The Role of TLPs in Leguminous Nodulation and Symbiosis

Nitrogen is a component of many important compounds in plants, and it plays a crucial role in the synthesis of proteins, nucleic acids, and chlorophylls. However, most plants are unable to directly use atmospheric nitrogen gas. Uniquely, legumes with special metabolic pathways can form root nodules with nitrogen-fixing bacteria to secure nitrogen for growth and development [83,84]. Soybean nodulation is controlled by several host genes referred to as *Rj* (*rj*) genes; these include the *Rj4* gene, which encodes a thaumatin-like protein. *Rj4* is constitutively expressed in roots, including in root nodules, so it is very likely that *Rj4* is involved in gene-for-gene resistance against specific Bradyrhizobium strains. These strains are highly competitive for nodulation but have low nitrogen fixation efficiency; thus, cultivars harboring an *Rj4* allele are considered favorable. It is also possible that Rj4 interacts with rhizobial surface polysaccharides given its glucan-binding and glucanase activities [57]. The nonmycorrhizal legume white lupin (*Lupinus albus*) prevents mycorrhiza infection, and a TLP peptide has been identified from its extracellular protein components [85]. TLPs have also been detected in the intercellular fluid and different tissues of healthy white lupins [86] and in the intercellular washing fluid of chickpea leaves [87], suggesting that these proteins may have inhibitory effects on symbiosis in addition to their anti-pathogenic functions.

In one study investigating short-term dynamics in proteomic patterns using continuous sampling, extrafloral nectar produced by an obligate ant plant (*Acacia cornigera*) showed that anabolism involving TLP accumulated in the plant’s nectary directly before secretion, and it diminished quantitatively after the daily secretion process [88]. This plant is inhabited by mutualistic ants, and large quantities of TLPs in the biochemically complex components secreted by its nectary may play a role in its symbiotic relationship with the insects.

### 3.5. Prediction of Protein–Protein Interactions in Legume TLPs

Protein–protein interactions in legume TLPs and the metabolic pathways they participate in may reflect their biological functions and potential roles. To demonstrate this, we analyzed protein–protein interactions between a TLP protein from *Arachis hypogaea*, AhTLP1 (NCBI accession number: XP_025651708.1) and two soybean TLP proteins, namely GmOLPa (NCBI accession number: NP_001236405.2) and GmOLPb (NCBI accession number: NP_001235877.1), using the STRING database (https://cn.string-db.org, accessed on 29 January 2024).

By analyzing the results using association rules and k-means clustering, we found that the interaction network of AhTLP1 can be divided into two clusters (Figure 3, Table 2). AhTLP1 may physically interact with 10 proteins, including the TYR_PHOSPHATASE_2 domain-containing protein, protein phosphatase 2C, the cyclic nucleotide-binding/kinase domain-containing protein, the PlsC domain-containing protein, 1-acyl-sn-glycerol-3-phosphate acyltransferase, 3-deoxy-d-manno-octulosonic-acid transferase, E3 ubiquitin–protein ligase, and two uncharacterized proteins. Functional association analyses showed that AhTLP1 may indirectly interact with another 10 proteins, including Rac-like GTP-binding protein RAC13, ribonucleoside–diphosphate reductase subunit beta, Clathrin heavy chain, the beta-adaptin-like protein, the FYVE-type domain-containing protein, the subunit of adaptor protein complex 2, and the beta-adaptin-like protein. 

KEGG pathway analysis has shown that the proteins of cluster 1 mainly participate in glycerolipid metabolism and glycerophospholipid metabolism because of the following two proteins: the PlsC domain-containing protein 1-acyl-sn-glycerol-3-phosphate acyltransferase (which is involved in the formation of phosphatidic acid), a precursor of various membrane phospholipids (PLs); and 3-deoxy-d-manno-octulosonic-acid transferase (involved in the biosynthesis of lipid A), a phosphorylated glycolipid that anchors the lipopolysaccharide to the outer membrane of the cell in bacteria [89]. The TYR_PHOSPHATASE_2 domain-containing protein is an enzyme responsible for dephosphorylating phosphor-Tyr from proteins [90]. Protein phosphatase type 2C functions in the ABA signaling pathway, which is one of the major signal transduction pathways in abiotic stress responses [91]. E3 ubiquitin–protein ligase transfers ubiquitin and ubiquitin-like proteins through an E2 enzyme to a target substrate, which is the final step of ubiquitination [92]. The AhTLP1’s interaction network indicates its potential role in different cellular processes. In addition, KEGG pathway analysis of cluster 2 has shown that there are 12 proteins involved in the endocytosis pathway. Importantly, one study showed that the thaumatin-like protein CalA in *Aspergillus fumigatus* assists this fungal pathogen in invading pulmonary epithelial cells and vascular endothelial cells by inducing its own endocytosis [93]. The above research shows that TLPs may play a role in the endocytosis pathway. 

For soybean TLP proteins, analyses of GmOLPa and GmOLPb protein–protein interactions conducted on the STRING database have shown that GmOLPa is involved in sucrose biosynthetic process peroxisome organization and the carbohydrate metabolic process (Figure 4A, Table 3). Furthermore, GmOLPb is involved in the sucrose biosynthetic process, defense responses to other organisms, the carbohydrate metabolic process, responses to biotic stimuli, and defense responses (Figure 4B, Table 4). 

Specifically, our protein–protein interaction analysis revealed that GmOLPa interacts with sucrose–phosphatase 2, Bet_v_1 domain-containing protein, regulatory protein NPR1, sucrose–phosphatase 1, ribonucleoprotein, the UBIQUITIN_CONJUGAT_2 domain-containing protein, calmodulin-like protein 1, and the ABC transporter family protein. Functional association analyses showed that GmOLPa may indirectly interact with another 10 proteins, including the glycosyl hydrolase 31 family, sucrose–phosphate synthase 2, and the peroxisome biogenesis proteins. Moreover, our protein–protein interaction analysis revealed that GmOLPb interacts with cytochrome P450 family proteins, trypsin inhibitor A, the Fe^2+^/Zn^2+^-regulated transporter, sucrose–phosphatase 1, and the bZIP transcription factor. The functional association analysis also showed that GmOLPb may indirectly interact with another 10 proteins, including Tim44 domain-containing proteins, sucrose–phosphate synthase 2, and the glycosyl hydrolase 31 family. Most importantly, the nonexpressor of pathogenesis-related genes 1 (NPR1) is a master regulator of SA-mediated SAR, a broad-spectrum disease-resistance mechanism in plants [94]. Basic leucine zipper (bZIP) genes encode transcription factors (TFs) that control important biochemical and physiological processes, including various kinds of abiotic and biotic stress responses, such as salt responses, drought responses, and pathogen responses in soybean [95,96,97]. This might explain why GmOLPs participate in high-salt and drought responses.

The above research shows that legume TLPs may be involved in multiple metabolic pathways. However, the proteins that interact with TLPs predicted by STRING must be confirmed through experimental verification to comprehensively explore their potential functions.

## 4. Potential Biotechnological Applications for Legume TLPs

The antifungal activity and stability of TLPs provide them with broad prospects for agricultural applications in the pathogen-resistance field. Highly conserved cysteine residues with disulfide bridge structures in most plant TLPs provide stabilization under extreme thermal and pH conditions, as well as resistance to protease degradation [98,99]. In addition, TLPs also protect plants from abiotic stresses, including cold, salinity, and drought [28,100]. These characteristics make TLPs a good source for breeding and plant transformation aimed at producing better performance under biotic and abiotic stresses.

Research shows that TLPs may play a role in plant herbicide resistance. Transcriptomic studies of chickpeas using two herbicide-susceptible and -tolerant genotypes exposed to imidazoline (Imazethapyr) have revealed that gene encoding for thaumatin-like protein-1 shows a five-fold change in differential expression as an effect of herbicide on tolerant plants, indicating that this TLP can be used for further investigation and association application studies [101].

Legume TLPs may have wide applications in the human food and pharmaceutical industries. Antifungal proteins (including TLPs) isolated from the seeds of legume plants have shown inhibitory activity against enzymes that are essential to the life cycle of human immunodeficiency virus type 1 (HIV-1); for instance, French bean TLPs have been found to potently inhibit HIV-1 integrase and reverse transcriptase, as well as low HIV-1 protease inhibitory activity [102]. Thaumatin has been approved and commercialized as a safe sweetener and flavor enhancer in food because it is 1600 times sweeter than sucrose on a weight-to-weight basis [103,104]. As homologies of thaumatin, TLPs can be developed as natural flavor modifiers or enhancers, replacing synthetic sweeteners. *TLP* genes can also be transferred to vegetables or fruit crops as flavor enhancers [28,105,106]. Exploring the pharmaceutical and sweet proteins in edible legume plants and their potential applications is a worthwhile research direction.

## 5. Perspectives

TLPs are a group of proteins with broad application prospects, but corresponding in-depth research on this family in leguminous plants remains quite limited. Many varieties of leguminous plants are edible and can be used for chemical industry purposes. Moreover, recombinant TLP products can be expressed and refined in different bioreactors, such as bacteria, fungi, and other plants, on account of the antifungal activity of TLPs. Furthermore, transgenic plants with TLPs have displayed increased activity under biotic and abiotic stresses. TLPs are valuable candidates for plant breeding. Further research on the precise mechanisms underlying legume TLP protein regulation, localization, and function is necessary, especially regarding whether TLPs can degrade fungal glucan, as well as whether they possess pathogen-associated molecular patterns for cell surface pattern recognition receptors, which can trigger plant immune responses in extracellular regions.

## Figures and Tables

**Figure 1 plants-13-01124-f001:**
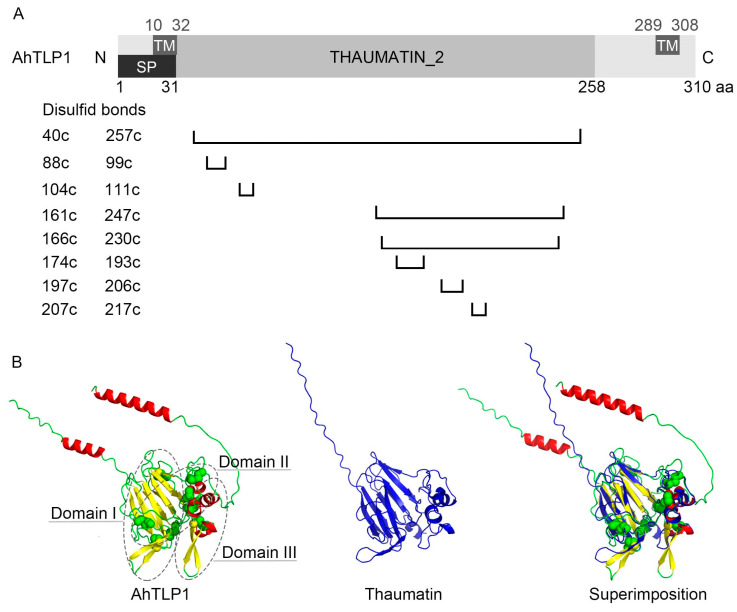
The protein structure of AhTLP1. (**A**) The protein structure and eight disulfide bonds of AhTLP (http://www.ebi.ac.uk./interpro/, accessed on 28 March 2024; http://prosite.expasy.org/, accessed on 28 March 2024). TM: transmembrane helix; SP: signal peptide; N: N-terminal; C: C-terminal; c: conserved cysteine involved in a disulfide bond. (**B**) Three-dimensional structure of AhTLP1 and thaumatin. The TLP protein family contains three conserved domains: domain I, containing 11 stranded β-sheets organized as a β-barrel, forming the protein core; domain II, containing an α-helix and a set of disulfide-rich loops (green balls); and domain III, containing a β-hairpin and a coil motif, both maintained by a disulfide bond. This image was generated with PyMOL Molecular Graphics System version 2.2.0 (Schrödinger, LLC, https://pymol.org).

**Figure 2 plants-13-01124-f002:**
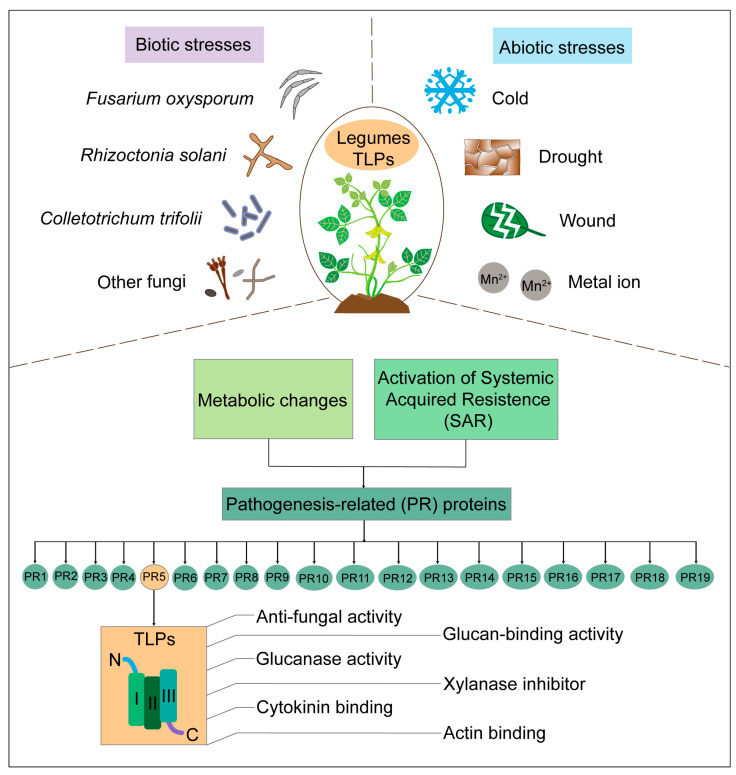
Plant defense mechanisms of TLPs. Plants respond to biotic and abiotic stresses through several pathways. Under stress conditions, metabolic changes and systemic acquired resistance (SAR) are quickly stimulated and initiate the expression of PR proteins, including PR5. TLPs are PR5 members, multifunctional proteins that exhibit antifungal activity, glucan-binding activity, glucanase activity, xylanase inhibitor abilities, cytokinin-binding abilities, and actin-binding abilities. TLPs contain three domains: I, II, and III. N: N-terminal, C: C-terminal.

**Figure 3 plants-13-01124-f003:**
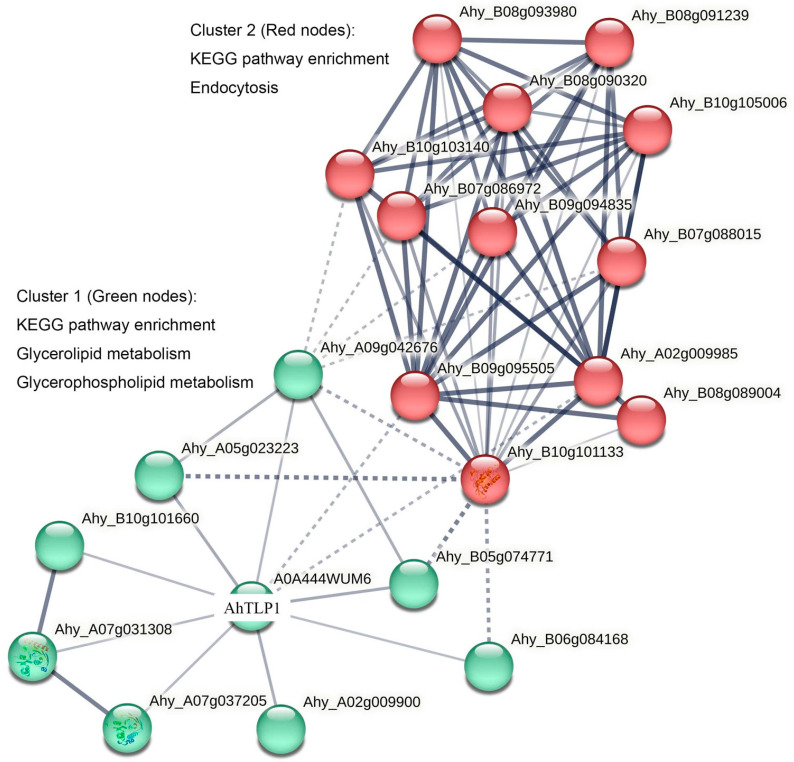
Protein–protein interaction predictions for *Arachis hypogaea* TLP1 analyzed using STRING. A0A444WUM6: AhTLP1, *Arachis hypogaea* TLP1; KEGG: Kyoto Encyclopedia of Genes and Genomes. The thickness of the line indicates the degree of confidence prediction of the interaction.

**Figure 4 plants-13-01124-f004:**
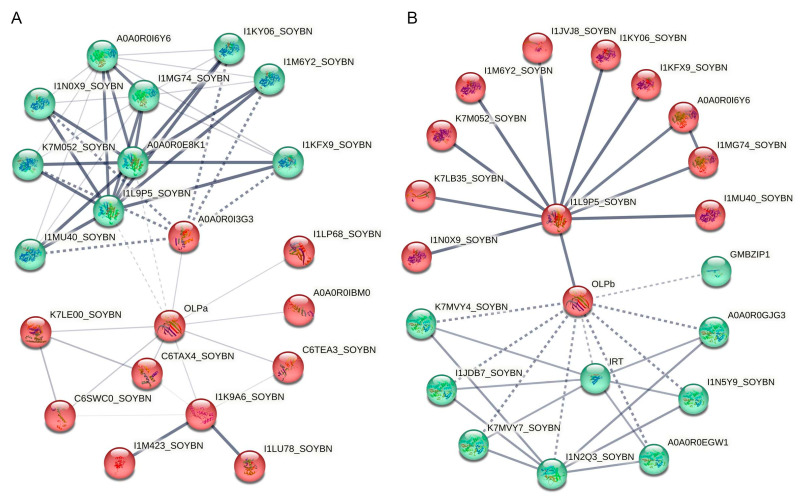
Protein–protein interaction predictions for *Glycine max* GmOLPa and GmOLPb analyzed using STRING. (**A**) STRING analysis of GmOLPa; (**B**) STRING analysis of GmOLPb. The thickness of the line indicates the degree of confidence prediction of the interaction.

**Table 2 plants-13-01124-t002:** The AhTLP1 protein–protein interacting network information.

Interactions	Predicted Interaction Partners	Protein Accessions
Direct interaction with AhTLP1	TYR_PHOSPHATASE_2 domain-containing protein	Ahy_A09g042676
	Protein phosphatase 2C and cyclic nucleotide-binding/kinase domain-containing protein	Ahy_A05g023223, Ahy_B05g074771
	PlsC domain-containing protein	Ahy_B10g101660, Ahy_A07g037205
	1-acyl-sn-glycerol-3-phosphate acyltransferase	Ahy_A07g031308
	3-deoxy-d-manno-octulosonic-acid transferase	Ahy_A02g009900
	E3 ubiquitin–protein ligase	Ahy_B06g084168
	Uncharacterized protein	Ahy_A02g009985, Ahy_B09g095505
Indirect interaction with AhTLP1	Rac-like GTP-binding protein RAC13	Ahy_B10g101133
	Ribonucleoside–diphosphate reductase subunit beta	Ahy_B08g089004
	Clathrin heavy chain	Ahy_B07g088015, Ahy_B09g094835, Ahy_B07g086972, Ahy_B10g103140
	Beta-adaptin-like protein	Ahy_B10g105006
	FYVE-type domain-containing protein	Ahy_B08g090320
	Subunit of the adaptor protein complex 2	Ahy_B08g093980
	Beta-adaptin-like protein	Ahy_B08g091239

**Table 3 plants-13-01124-t003:** GmOLPa protein–protein interaction network information.

Interactions	Predicted Interaction Partners	Protein Accessions
Direct interaction with GmOLPa	Sucrose–phosphatase 2	A0A0R0I3G3, A0A0R0E8K1
	Bet_v_1 domain-containing protein	I1LP68_SOYBN
	Regulatory protein NPR1	A0A0R0IBM0
	Sucrose–phosphatase 1	I1L9P5_SOYBN
	Ribonucleoprotein	C6TEA3_SOYBN
	UBIQUITIN_CONJUGAT_2 domain-containing protein	I1K9A6_SOYBN
	Calmodulin-like protein 1	C6SWC0_SOYBN, C6TAX4_SOYBN
	ABC transporter family protein	K7LE00_SOYBN
Indirect interaction with GmOLPa	Glycosyl hydrolase 31 family	I1MG74_SOYBN, A0A0R0I6Y6
	Sucrose–phosphate synthase 2	I1KY06_SOYBN, I1M6Y2_SOYBN, I1N0X9_SOYBN, I1KFX9_SOYBN, K7M052_SOYBN, I1MU40_SOYBN
	Peroxisome biogenesis protein	I1M423_SOYBN, I1LU78_SOYBN

**Table 4 plants-13-01124-t004:** GmOLPb protein–protein interaction network information.

Interactions	Predicted Interaction Partners	Protein Accessions
Direct interaction with GmOLPb	Cytochrome P450 family protein	K7MVY7_SOYBN, A0A0R0GJG3, I1N5Y9_SOYBN, K7MVY4_SOYBN, A0A0R0EGW1, I1JDB7_SOYBN
	Trypsin inhibitor A	I1N2Q3_SOYBN
	Fe^2+^/Zn^2+^ regulated transporter	IRT
	Sucrose–phosphatase 1	I1L9P5_SOYBN
	bZIP transcription factor	GmbZIP1
Indirect interaction with GmOLPb	Tim44 domain-containing protein	I1JVJ8_SOYBN, K7LB35_SOYBN
	Sucrose–phosphate synthase 2	I1KY06_SOYBN, I1KFX9_SOYBN, I1M6Y2_SOYBN, K7M052_SOYBN, I1N0X9_SOYBN, I1MU40_SOYBN
	Glycosyl hydrolase 31 family	A0A0R0I6Y6, I1MG74_SOYBN

## Data Availability

No new data were created or analyzed in this study. Data sharing does not apply to this article.

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
