# Peer review of "Thaumatin-like Proteins in Legumes: Functions and Potential Applications—A Review"

_plants, 2024, doi:10.3390/plants13081124_

Round 1

Reviewer 1 Report

Comments and Suggestions for Authors

Thaumatin-like Proteins in Legumes: Functions and Potential Applications – A Review

Lanlan Feng, Shaowei Wei, and Yin Li

     The review of Feng et al. is dedicated to a group of proteins that belong to the group of Thaumatin-like proteins (TLPs). Feng et al.  analyzed and described the data concerning the classification and the  structure of TLPs, as well as the role they are playing in response to biotic and abiotic stress.

     Authors have selected for the review the analysis of TLPs in legume plants. Taking into account the role of legumes in soil fertility, as well as legume products which are used for food and other purposes, the selection of the topic is justified.  Feng et al. have reviewed a large amount of literature and have conducted the analysis of protein-protein interactions using available bioinformatic resources ( sub-chapter 3.5).     

  However, the text needs to be revised as it is sometimes difficult to understand and gives the impression to be somehow chaotic. I added quite a number of sticky notes to PDF version of the paper (attached). I think it’s best to rework the text with an English-language editor.

I would recommend shortening Section 2, since the data are presented in Table 1, and give more detailed explanation of biological role of TLPs proteins that are referred in sub-chapter.

The data from sub-chapted 3.5  better to be presented as tables.

I would advise the Authors to restructure the review and provide a short text before each sub-chapter that would explain the biological mechanisms in which the proteins described in this text are involved. Simply listing the data, as currently presented, is not optimal for the reader who is not directly involved in the topic of TLPs proteins.

The paper needs major revision.

Comments on the Quality of English Language

The text must be revised by an English-language editor

Reviewer 2 Report

Comments and Suggestions for Authors

The manuscript is a comprehensive review of the TLP proteins in Legume, which covers various aspects such as the classification, structure, and host resistance to biotic and abiotic stresses. The authors also analyzed possible protein-protein interactions and further discussed the significance of investigating the function and mechanism of TLP proteins for agricultural and biomedical applications.

I didn’t find major flaws of this review but have a few suggestions.

1)        I don’t see the necessity of Figure 1. It can be combined with Figure 2 for a concise writing and figure planning.

2)        If I understood correctly, Figure 3 and 4 are similarly analysis for different proteins, but the figure legends are too brief to tell what the difference between the rainbow colored lines (in Figure 4)  vs. the solid black lines in Figure 3. Please add more explanations or keep the same formatting when generating the figures.

3)        I wound suggest adding figures of the structure of the proteins (lines 101-106) and the sub cellular localization (lines 108-118). That is actually more interesting to readers than a general illustration like Figure 1.

4)        The logics of Introduction would need to improve. Lines 36-45 discussed the abiotic stress of plants in general, and Lines 46-57 are about the biotic stress of legume, and the last paragraph (Lines 59-80) went back to talk about mainly on the pathogen infection (part of the biotic stress) and the role of TLPs. But this paper is about TLPs and their roles in defensing both biotic and abiotic stresses in beans. The logic flow is not optimal. Please rewrite so that it is easier to follow.

5)        Minor suggestions: please use the scientific name of the plants if you can, like “soybeans” or “wild peanuts” (lines 97-98) should have their Latin names to follow the proper taxonomy. You could use both common names and their scientific names.

6)        Minor suggestions: line 133: repetitive of biological stress and biotic stress; line 137, please explain the abbreviation Cd.

Comments on the Quality of English Language

The English is good.

Round 2

Reviewer 1 Report

Comments and Suggestions for Authors

Authors made a good job. The review has been significantly   improved, and now it looks nice and interesting. The review can be accepted for publication.